# *PKD1* Nonsense Variant in a Lagotto Romagnolo Family with Polycystic Kidney Disease

**DOI:** 10.3390/genes14061210

**Published:** 2023-06-01

**Authors:** Michaela Drögemüller, Nadine Klein, Rikke Lill Steffensen, Miriam Keiner, Vidhya Jagannathan, Tosso Leeb

**Affiliations:** 1Institute of Genetics, Vetsuisse Faculty, University of Bern, 3001 Bern, Switzerland; michaela.droegemueller@unibe.ch (M.D.); vidhya.jagannathan@unibe.ch (V.J.); 2Tierärztliche Praxis für Kleintiere, Dickstrasse 57, 53773 Hennef (Sieg), Germany; nadine_klein@icloud.com; 3Schlitterweg 6, 61191 Rosbach, Germany; rikkelill@yahoo.dk; 4Small Animal Clinic, Internal Medicine, Justus-Liebig-University, 35392 Giessen, Germany; miriam.keiner@icloud.com

**Keywords:** Canis lupus familiaris, dog, whole genome sequencing, de novo, precision medicine, HRFCD, ADPKD, animal model

## Abstract

A female Lagotto Romagnolo dog with polycystic kidney disease (PKD) and her progeny, including PKD-affected offspring, were studied. All affected dogs appeared clinically inconspicuous, while sonography revealed the presence of renal cysts. The PKD-affected index female was used for breeding and produced two litters with six affected offspring of both sexes and seven unaffected offspring. The pedigrees suggested an autosomal dominant mode of inheritance of the trait. A trio whole genome sequencing analysis of the index female and her unaffected parents identified a de novo heterozygous nonsense variant in the coding region of the *PKD1* gene. This variant, NM_001006650.1:c.7195G>T, is predicted to truncate 44% of the open reading frame of the wild-type PKD1 protein, NP_001006651.1:p.(Glu2399*). The finding of a de novo variant in an excellent functional candidate gene strongly suggests that the *PKD1* nonsense variant caused the observed phenotype in the affected dogs. Perfect co-segregation of the mutant allele with the PKD phenotype in two litters supports the hypothesized causality. To the best of our knowledge, this is the second description of a *PKD1*-related canine form of autosomal dominant PKD that may serve as an animal model for similar hepatorenal fibrocystic disorders in humans.

## 1. Introduction

Hepatorenal fibrocystic disorders (HRFCDs) are characterized by developmental portobiliary and renal fibrocystic abnormalities, such as polycystic kidneys and congenital hepatic fibrosis [1]. HRFCDs belong to a larger group of disorders called ciliopathies, which are caused by structural or functional defects in the primary cilium [2]. Autosomal recessive and dominant polycystic kidney disease (ARPKD and ADPKD, respectively) are the most common forms of life-threatening monogenic HRFCD in humans [3]. ADPKD is one of the most common inherited disorders known in humans and is characterized by multiple cysts in both kidneys that often lead to chronic and end-stage renal disease. It may be caused by pathogenic variants in *PKD1* encoding polycystin 1, transient receptor potential channel interacting (OMIM 173900), or *PKD2* encoding polycystin 2, transient receptor potential channel interacting (OMIM 173910) [3]. Polycystin 1 and 2 localize to the cilia of renal epithelial cells, and their function is thought to involve an inhibitory activity that suppresses the cilia-dependent cyst activation (CDCA) signal. Polycystin deficiency results in the activation of CDCA and stimulation of cyst growth [2]. Although the genotype–phenotype correlation in ADPKD is not fully understood; in human patients, it has been reported that renal survival associated with *PKD2* variants is approximately 20 years longer than that associated with *PKD1* variants [4]. Animal models have played an important role in the understanding of disease onset and progression and in the development of therapeutic interventions [5].

Spontaneous animal models, mostly rodents, have long been used to study human diseases. However, the key pathological processes underlying cyst development and exacerbation in pre-symptomatic stages remain unknown because rodent models do not recapitulate critical disease phenotypes, including disease onset in heterozygotes. With the knowledge of the relevant genes, targeted genetically modified models have been established. ADPKD models with *PKD1* variants leading to severe cyst formation have been generated first in minipigs (OMIA 000807-9823) and more recently in cynomolgus macaques (OMIA 000807-9541) [6,7]. Spontaneous animal models have also been reported in domestic animals, first in Persian cats (OMIA 000807-9685), in which a widespread *PKD1*:p.Cys3294* nonsense variant was discovered as a cause for a feline form of ADPKD in 2004 [8]. In 2011, a missense variant in the canine *PKD1* gene, p.Glu3187Lys, was described as the cause of ADPKD in an Australian Bull Terrier family (OMIA 000807-9615) [9]. A distinct form of ADPKD has been reported in a Siberian cat with a frameshift variant in the *PKD2* gene (OMIA 002525-9685) [10].

Further documented instances of canine HRFCDs include older reports on Cairn Terriers and West Highland White Terriers [11,12]. An autosomal recessive form of HRFCD in Norwich Terriers is caused by a splice site variant in the *INPP5E* gene encoding inositol polyphosphate-5-phosphatase E (OMIA:002173-9615) [13].

The aim of this study was to characterize the phenotype and identify the underlying genetic variant in a family of purebred Lagotto Romagnolo dogs segregating an ADPKD.

## 2. Materials and Methods

### 2.1. Clinical Examination and Investigations

Two offspring of the same female (index female) had independent abdominal ultrasound examinations at six and seven months of age, respectively, due to medical problems not related to this study. Incidental findings of cysts in the kidneys of both dogs prompted the breeder to initiate transabdominal ultrasound examinations in all close relatives. This included 11 additional littermates (examined between 12 and 18 months of age), the dam (examined at 3 years of age), 3 sires (examined at ≥3 years of age), and the female’s parents (examined at ≥5 years of age).

### 2.2. Animals and DNA Extraction

Blood samples from the index female, 4 of her littermates, her parents, the 3 sires of both litters, one of which was planned dual-sired, and 12 puppies, including 6 affected, were available and used for DNA extraction. The parentages of the puppies in the dual-sired litter were established by DNA profiles (Laboklin, Bad Kissingen, Germany). In addition to the 22 family members, 129 blood samples from unrelated Lagotto Romagnolo dogs, which had been donated to the Vetsuisse Biobank, were used. Genomic DNA was isolated from EDTA blood using the Maxwell RSC Whole Blood Kit on a Maxwell RSC instrument (Promega, Dübendorf, Switzerland).

### 2.3. Whole Genome Sequencing (WGS)

Three Illumina TruSeq PCR-free DNA libraries of ~400 bp insert size were generated from the affected index female and her unaffected parents. We collected 180 (index female), 194 (sire), and 187 (dam) million 2 × 150 bp paired-end reads on a NovaSeq 6000 instrument. Mapping to the UU_Cfam_GSD_1.0 reference genome assembly was performed as described [14], resulting in 20-22x average sequence depth. The sequence data were deposited in the European Nucleotide Archive under study accession number PRJEB16012 and sample accession numbers SAMEA110415694 (affected index female), SAMEA110415696 (sire), and SAMEA110415695 (dam).

### 2.4. Variant Calling

Variant calling was performed using GATK HaplotypeCaller [15] in gVCF mode as described [14]. To predict the functional effects of the called variants, the SnpEff [16] software, together with the UU_Cfam_GSD_1.0 reference genome assembly and NCBI annotation release 106, was used. For variant filtering, we used 958 control dog genomes from different breeds. The control genomes comprised 27 Lagotto Romagnolo dogs, including a previously studied dog with *COL1A2*-related osteogenesis imperfecta that was a littermate of the PKD-affected index female studied here [17]. Accessions of all genome sequences are listed in Appendix A.

### 2.5. Gene Analysis

Numbering within the canine *PKD1* gene corresponds to the NCBI RefSeq accession numbers NM_001006650.1 (mRNA) and NP_001006651.1 (protein).

### 2.6. Allele-Specific PCR and Sanger Sequencing

Primers 5′-GGG AGA CGC ACA GGT GAT-3′ (Primer F) and 5′-CTC ACC CCT CGC TTG GAA -3′ (Primer R) were used to generate an amplicon containing the *PKD1*:c.7195G>T variant. PCR products were amplified from genomic DNA using AmpliTaq Gold 360 Master Mix (Thermo Fisher Scientific, Reinach, Switzerland). Direct Sanger sequencing of the PCR amplicons was performed on an ABI 3730 DNA Analyzer (Thermo Fisher Scientific, Reinach, Switzerland) after treatment with exonuclease I and alkaline phosphatase. Sanger sequences were analyzed using the Sequencher 5.1 software (Gene Codes, Ann Arbor, MI, USA).

## 3. Results

### 3.1. Clinical Phenotype

Following the incidental discovery of renal cysts in two offspring of a Lagotto Romagnolo female, transabdominal ultrasound examinations were performed on 19 members of the entire family, consisting of the index female, offspring of her two litters, the litters’ three sires, and the index female’s parents. These examinations revealed the presence of renal cysts in the index female and six of her offspring (Figure 1). Two of the affected offspring were additionally diagnosed with liver cysts. The kidneys of the index female’s parents and the litters’ sires were clinically unremarkable. At the time of the investigations, the dogs with kidney cysts did not show any clinically overt phenotype. The ultrasound findings were diagnosed as polycystic kidney disease (PKD).

### 3.2. Pedigree Analysis and Mode of Inheritance

The index female and 6 of her 13 offspring were diagnosed as PKD-affected. The affected offspring included five male and one female dog. The parents of the index female, the three sires of the two litters, and the seven remaining offspring of the index female were unaffected. This pedigree, with roughly half of the index female’s offspring being affected, suggested autosomal dominant inheritance of the trait in the offspring of the index female. As the index female’s parents were both unaffected, we hypothesized that a de novo mutation event in the germline of one of her parents or during the early embryonic development of the index female generated a new pathogenic allele. The parentage of the index female’s sire and dam was verified by the WGS of the trio (see below).

### 3.3. Whole Genome Sequencing

We sequenced the genome of the index female and her parents and compared the data to 958 other control genomes from genetically diverse dogs. We performed a trio analysis, comparing the genotypes of the PKD-affected index female with the genotypes of both unaffected parents and simultaneously excluded variants that were also present in any of the 958 control genomes. We considered two alternative scenarios for the putative causal variant: for an autosomal recessive trait, we expected the affected female to be homozygous for the alternative allele and both parents to be heterozygous. Conversely, for an autosomal dominant trait caused by a de novo mutation event, the affected female should be heterozygous, and both parents homozygous for the reference allele. Table 1 summarizes the results of the variant filtering. Additional details of the private protein-changing variants are listed in Appendix A.

Of the four identified candidate variants for which the parental genotypes were compatible with a pathogenic effect (Appendix A), only one resided in a known candidate gene for renal cysts. The variant was a heterozygous nonsense variant in exon 17 of *PKD1*, an excellent functional candidate gene for ADPKD. The *PKD1* variant can be designated as chr6:39,295,382G>T (UU_Cfam_GSD_1.0 assembly; Figure 2a) or NM_001006650.1:c.7195G>T. The variant designation for the predicted protein is NP_001006651.1:p.(Glu2399*). It is unclear whether the mutant protein is actually expressed. Since the premature stop codon is predicted to truncate more than 44% of the 4311 codons of the wildtype open reading frame, it is very unlikely that a potentially expressed mutant protein has any physiological function.

The trio analysis comparing the variants in the affected dog with the genomes of both parents confirmed our hypothesis that NM_001006650.1:c.7195G>T has arisen by a de novo mutation event, as the mutant allele was absent from the leukocyte DNA of both parents (Figure 2a). The presence of the variant in the affected dog and its absence in both parental genomes was confirmed by Sanger sequencing (Figure 2b). Six additional affected dogs in the progeny of the sequenced PKD-affected female also carried the mutant allele in a heterozygous state (Figure 2c). In addition, 129 unrelated Lagotto Romagnolo dogs were genotyped and found to be clear of the mutant allele. The variant was also absent in 931 sequenced genomes from other dog breeds (Table 2).

## 4. Discussion

We identified a *PKD1* nonsense variant, NP_001006651.1:p.(Glu2399*), in a canine family with ADPKD. Nonsense variants are generally assumed to result in a complete loss of function in accordance with established guidelines for the interpretation of sequence variants in human medicine [19]. Experimental confirmation that the variant has arisen by a de novo mutation event, co-segregation of the mutant allele with the ADPKD phenotype in a moderately sized family, and absence of the mutant allele from a large cohort of dogs allow to classify this variant as pathogenic according to human diagnostic criteria [19] and establish it as the cause of the observed renal changes. To the best of our knowledge, this is the second reported pathogenic *PKD1* variant in dogs after the report of PKD1:pE3187K in Bull Terriers with ADPKD [9].

Polycystin 1, encoded by the *PKD1* gene, forms a complex with polycystin 2 that regulates multiple signaling pathways to maintain normal renal tubular structure and function [20]. Data from large human genome sequencing studies presented in the Genome Aggregation Database (gnomAD) [21] showed that the pLI score, the probability of loss-of-function intolerance, for *PKD1*, is 1, which means that *PKD1* belongs to the class of loss-of-function haploinsufficient genes, where there is no tolerance for loss-of-function variants in one copy of a gene. Therefore, the nonsense variant detected here is predicted to cause haploinsufficiency. Overall, numerous studies indicate that monoallelic variants in *PKD1* may lead to ADPKD in different species. Human *PKD1* variants that result in haploinsufficiency have been reported to cause ADPKD (OMIM 601313) [22,23]. Truncating *PKD1* variants are correlated with an earlier age of disease onset than non-truncating variants [4]. In addition, a truncating variant in the feline *PKD1* ortholog is also responsible for ADPKD in Persian cats [8]. To investigate the hypothesized reduction in expression and haploinsufficiency as the primary disease mechanism, mRNA, or protein expression studies in affected tissue samples would have been useful. Abnormalities in the polycystins lead to aberrant signaling through the downstream signaling pathways [24]. However, because the expression is at its highest in fetal kidney tissue and at its lowest in adult tissue [25,26], appropriate samples for such an analysis were unfortunately not available for the investigated family of dogs.

Recently, the re-expression of murine polycystins in established cysts has been shown to be effective in the reversal of PKD in mice [27], suggesting that ADPKD may be cured in the future. These findings indicated that the polycystins actively mediate luminal contraction. In this way, the function of the polycystins is similar to that of a “cruise control” to keep the diameter of the lumen within a certain range [2].

As far as we know, all reported cases of PKD in Lagotto Romagnolo dogs to date descended from the PKD-affected index female studied here, making it very likely that there is currently only one disease-causing variant in the breed. However, as shown here, new mutations can occur. They may explain similar familial forms of ADPKD in the future. If an offspring of the PKD-affected female is to be used for breeding, genetic testing should be performed to avoid unintentional breeding with affected dogs to prevent the spread of this deleterious allele. Since truncating *PKD1* variants lead to progressive renal failure in humans and cats, it, unfortunately, seems likely that the life expectancy of the ADPKD-affected Lagotto Romagnolo dogs reported herein will be reduced.

Interestingly, another de novo mutation event in the *COL1A2* gene caused a severe form of osteogenesis imperfecta in a brother of the PKD-affected index female [17]. To the best of our knowledge, this is the first report of canine littermates with two independent disease-causing de novo variants. It is not known whether the parents of the two affected dogs have been exposed to any mutagenic events.

## 5. Conclusions

Based on the clinical and genetic findings, ADPKD was diagnosed in a family of Lagotto Romagnolo dogs. We identified a *PKD1* nonsense variant as the likely cause and demonstrated that it had arisen de novo either in the germline of one of the parents or during the early embryonic development of the affected index female. The identified dogs represent an animal model to improve our understanding of ADPKD and potentially for the development of future therapeutic strategies.

## Figures and Tables

**Figure 1 genes-14-01210-f001:**
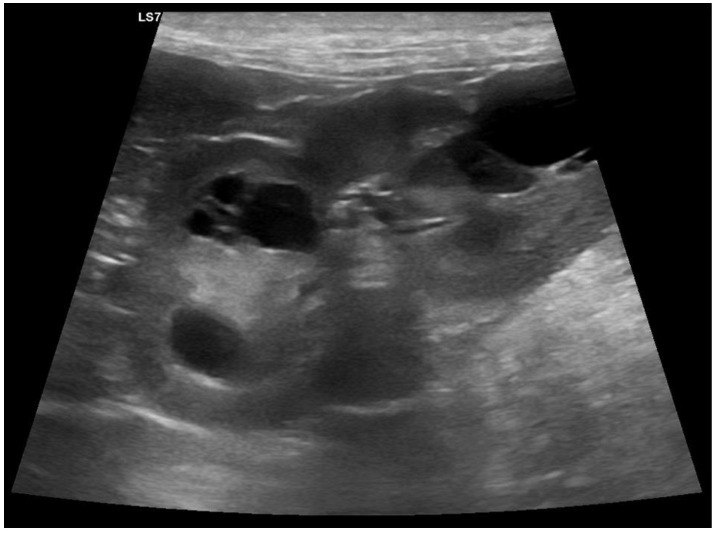
Ultrasound examination of the left kidney of the affected index female. Multiple cysts are visible.

**Figure 2 genes-14-01210-f002:**
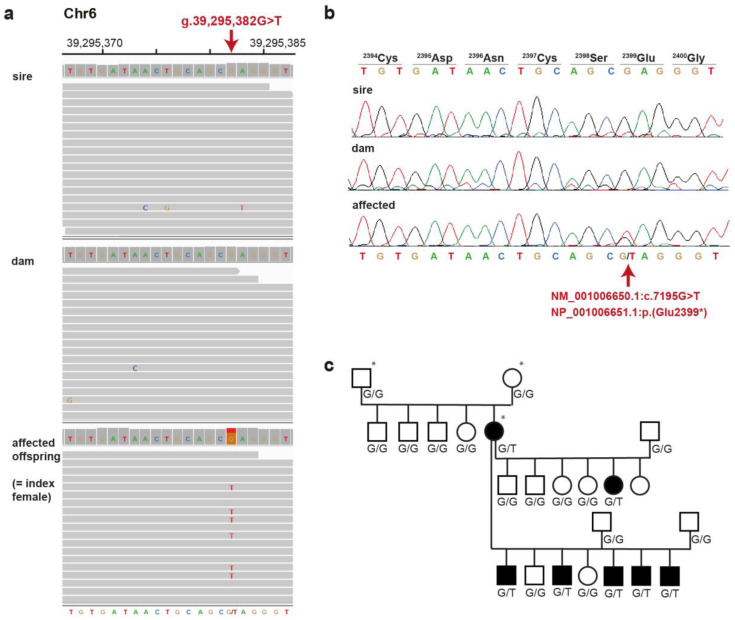
Details of the NM_001006650.1:c.7195G>T, NP_001006651.1:p.(Glu2399*) variant. (**a**) IGV [18] screenshot of the PKD-affected index female and her parents illustrating the de novo mutation event. (**b**) Sanger sequencing confirmed the NM_001006650.1:c.7195G>T variant, which alters codon 2399, predicted to result in a premature stop codon. (**c**) Pedigree showing the perfect co-segregation of the genotypes at the *PKD1*:c.7195G>T variant with the ADPKD phenotype. Black symbols indicate ADPKD-affected dogs; the WGS trio is indicated by asterisks. Please note that the index female was mated to two different sires for her second litter.

**Table 1 genes-14-01210-t001:** Variants detected by whole genome sequencing of the affected Lagotto Romagnolo female.

Filtering Step	Heterozygous Variants	Homozygous Variants
Variants in the whole genome	3,608,700	2,610,423
Private protein-changing variants	2	2 ^a^

^a^ Heterozygous carriers among the 27 Lagotto Romagnolo dogs with WGS were allowed.

**Table 2 genes-14-01210-t002:** Association of the genotypes at NM_001006650.1:c.7195G>T with the ADPKD phenotype.

	GG	GT	TT
PKD-affected index female	-	1	-
Unaffected parents of the PKD-affected index female	2	-	-
PKD-affected offspring of the index female	-	6	-
Unaffected related dogs (offspring and littermates of the index female)	13	-	-
Other Lagotto Romagnolo dogs ^a^	103	-	-
Sequenced Lagotto Romagnolo genomes ^a^	26	-	-
Sequenced dog genomes from various other breeds ^a^	931	-	-

^a^ Phenotypes are unknown.

## Data Availability

The accession numbers of the sequence data that are reported in this study are listed in Appendix A.

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
