# Peer review of "PKD1 Nonsense Variant in a Lagotto Romagnolo Family with Polycystic Kidney Disease"

_genes, 2023, doi:10.3390/genes14061210_

Round 1

Reviewer 1 Report

Overview

The manuscript by Drögemüller and collaborators focuses on the identification of a new DNA variant for autosomal dominant polycystic kidney disease (ADPKD) in dogs. Authors recruited a canine family segregating ADPKD and performed a trio whole genome sequencing analysis and variant filtering. They identified a de novo nonsense variant.

The paper is clearly written and straightforward, identifying a variant in the gene PKD1, one of the most common gene defects responsible for ADPKD in humans and animals. This work will certainly be appreciated by the dog community of breeders, veterinarians and geneticists from the field.

Specific comments

Please homogenize the designation of the variant in the abstract and along the manuscript and tables, at both DNA and protein levels:

NM_001006650.1:c.7195G>T or PKD1:c.7195G>T

NP_001006651.1:p.(Glu2399*) or PKD1:p.(Glu2399*)

Introduction

Please add the PKD2 variant found in a Siberian cat (Rodney et al., 2021).

Materials and Methods

2.5. Gene Analysis: the reference for the PKD1 protein is not the right one: XP_038397573.1 refers to ACADM protein.

Results

3.1. Clinical phenotype: dogs from the family were screened by ultrasound examination but the clinical description would have benefited some biochemical data. How were renal biochemical parameters (urea, creatinine...) from affected dogs?

Another data is lacking: ages of the first two puppies that were screened is mentioned but how old were the adult dogs (index female, 3 sires and parents) when they were screened?

3.3. Whole genome sequencing

L155: please add a short description of the other 3 candidate variants.

L162: please be cautious and change "truncates" by "is predicted to truncate".

Discussion

In the first paragraph please add some discussion about the three other candidate variants.

L225: discussion about life expectancy of affected dogs should mention the age of affected dogs at ultrasound diagnosis and why no clinical phenotype was observed.

Author Response

(1)

The manuscript by Drögemüller and collaborators focuses on the identification of a new DNA variant for autosomal dominant polycystic kidney disease (ADPKD) in dogs. Authors recruited a canine family segregating ADPKD and performed a trio whole genome sequencing analysis and variant filtering. They identified a de novo nonsense variant.

The paper is clearly written and straightforward, identifying a variant in the gene PKD1, one of the most common gene defects responsible for ADPKD in humans and animals. This work will certainly be appreciated by the dog community of breeders, veterinarians and geneticists from the field.

Response: Thank you very much for the compliments and for the many excellent specific comments that greatly helped to improve the manuscript.

(2)

Please homogenize the designation of the variant in the abstract and along the manuscript and tables, at both DNA and protein levels:

NM_001006650.1:c.7195G>T or PKD1:c.7195G>T

NP_001006651.1:p.(Glu2399*) or PKD1:p.(Glu2399*)

Response: We revised the manuscript accordingly and use now exclusively the HGVS approved variant designations including the accession numbers for the reference sequences.

(3)

Introduction

Please add the PKD2 variant found in a Siberian cat (Rodney et al., 2021).

Response: We added this reference to the introduction as requested.

(4)

Materials and Methods

2.5. Gene Analysis: the reference for the PKD1 protein is not the right one: XP_038397573.1 refers to ACADM protein.

Response: Thank you very much for spotting this error! We corrected it accordingly.

(5)

Results

3.1. Clinical phenotype: dogs from the family were screened by ultrasound examination but the clinical description would have benefited some biochemical data. How were renal biochemical parameters (urea, creatinine...) from affected dogs?

Another data is lacking: ages of the first two puppies that were screened is mentioned but how old were the adult dogs (index female, 3 sires and parents) when they were screened?

Response: The reviewer raises an important question. We obtained a few biochemical test results, but these were not taken consistently and not from all affected dogs in the study. The available biochemical parameters were within norms (kidney and liver). However, as we consider the available data insufficient for a scientific publication, we did not include them in our manuscript. This should be done in a separate longitudinal study that will monitor the development of these values during the lifetime of the affected dogs.

We added the age of the dogs at the time of the ultrasound examinations to the Methods section (lines 77-79).

(6)

3.3. Whole genome sequencing

L155: please add a short description of the other 3 candidate variants.

Response: The other 3 variants reside in LPCAT4 encoding lysophosphatidylcholine acyltransferase 4, NUP210L encoding "nucleoporin 210 like" and LOC100687820. The function of the latter two genes is unknown. For LPCAT4, an enzymatic activity has been described. However, no report on a possible phenotype in human patients or animal mutants is available. We think that a verbose description of these three genes would disrupt the flow of the main text too much. To address the comment, we added additional information to Table S2 and added another reference to Table S2 in line 156, so that an interested reader can easily find this information.

(7)

L162: please be cautious and change "truncates" by "is predicted to truncate".

Response: We changed revised the statement as requested.

Reviewer 2 Report

The authors present an interesting study on familiar polycystic kidney disease in dogs. The mode of inheritance and underlying genetic alterations are well described.This article is a valuable contribution to polycystic kidney disease in dogs.

As a pathologists I can only partially comment on the article. Review by a geneticist should be requested to get feedback on the applied methods. Pathologic examination of the affected organs (which would be my field of expertise) were not part of the study; however, it seems appropriate to me to diagnose the disease based on the sonographic findings, pared with the mutation analysis and affection of multiple family members. Below is a somewhat extended review. The article entitled "PKD1 nonsense variant in a Lagotto Romagnolo family with 2 polycystic kidney disease" describes a rare inherited disease in a bitch and her offsprings. Polycystic kidney disease is currently very rarely described in dogs. There is a lack of literature on this disease in dogs and this article provides relevant new insights. The authors have identified a specific mutation that was associated with the affected dogs, which is highly valuable for development of genetic tests for dogs and for comparison to other species including humans. The methodology seems appropriate to me; however, I am not a geneticist and I suggest that a reviewer with better knowledge on genetics should comment on the mutation analysis and pedigree development. I would like the authors to comment if cysts were found in other organs (such as liver and pancreas). The conclusions are supported by the results and the used references are appropriate. The tables and figures nicely supports understanding the main findings of the article

Author Response

The authors present an interesting study on familiar polycystic kidney disease in dogs. The mode of inheritance and underlying genetic alterations are well described.This article is a valuable contribution to polycystic kidney disease in dogs.

As a pathologists I can only partially comment on the article. Review by a geneticist should be requested to get feedback on the applied methods. Pathologic examination of the affected organs (which would be my field of expertise) were not part of the study; however, it seems appropriate to me to diagnose the disease based on the sonographic findings, pared with the mutation analysis and affection of multiple family members. Below is a somewhat extended review. The article entitled "PKD1 nonsense variant in a Lagotto Romagnolo family with 2 polycystic kidney disease" describes a rare inherited disease in a bitch and her offsprings. Polycystic kidney disease is currently very rarely described in dogs. There is a lack of literature on this disease in dogs and this article provides relevant new insights. The authors have identified a specific mutation that was associated with the affected dogs, which is highly valuable for development of genetic tests for dogs and for comparison to other species including humans. The methodology seems appropriate to me; however, I am not a geneticist and I suggest that a reviewer with better knowledge on genetics should comment on the mutation analysis and pedigree development. I would like the authors to comment if cysts were found in other organs (such as liver and pancreas). The conclusions are supported by the results and the used references are appropriate. The tables and figures nicely supports understanding the main findings of the article.

Response: We thank the reviewer for these very kind comments. As to the question of cysts in other organs, we stated that two of the affected dogs had liver cysts in addition to their kidney cysts (line 126).